# Jeju Citrus (*Citrus unshiu*) Leaf Extract and Hesperidin Inhibit Small Intestinal α-Glucosidase Activities *In Vitro* and Postprandial Hyperglycemia in Animal Model

**DOI:** 10.3390/ijms252413721

**Published:** 2024-12-23

**Authors:** Gi-Jung Kim, Yelim Jang, Kyoung-Tae Kwon, Jae-Won Kim, Seong-IL Kang, Hee-Chul Ko, Jung-Yun Lee, Emmanouil Apostolidis, Young-In Kwon

**Affiliations:** 1Department of Food and Nutrition, Hannam University, Daejeon 34054, Republic of Korea; homina97@daum.net (G.-J.K.); yelim_0808@naver.com (Y.J.); hangarambng@hangarambnf.com (K.-T.K.); seembeeks@hanmail.net (J.-Y.L.); 2Jeju Institute of Korean Medicine, Jujusi, Juju 63309, Republic of Korea; kjw8839@jikom.or.kr (J.-W.K.); sikang@jikom.or.kr (S.-I.K.); ifly1007@jikom.or.kr (H.-C.K.); 3Department Chemistry and Food Science, Framingham State University, Framingham, MA 01701, USA

**Keywords:** α-glucosidases, postprandial hyperglycemia, *Citrus unshiu* leaf, diabetes, inhibition

## Abstract

Citrus fruits are widely distributed in East Asia, and tea made from citrus peels has demonstrated health benefits, such as a reduction in fever, inflammation, and high blood pressure. However, citrus leaves have not been evaluated extensively for their possible health benefits. In this study, the α-glucosidase-inhibitory activity of Jeju citrus hot-water (CW) and ethyl alcohol (CE) extracts, along with hesperidin (HP) (a bioactive compound in citrus leaf extracts), was investigated, and furthermore, their effect on postprandial blood glucose reduction in an animal model was determined. The hesperidin contents of CW and CE were 15.80 ± 0.18 and 39.17 ± 0.07 mg/g-extract, respectively. Hesperidin inhibited α-glucosidase (IC_50_, 4.39), sucrase (0.50), and CE (2.62) and demonstrated higher α-glucosidase inhibitory activity when compared to CW (4.99 mg/mL). When using an SD rat model, during sucrose and starch loading tests with CE (*p* < 0.01) and HP (*p* < 0.01), a significant postprandial blood glucose reduction effect was observed when compared to the control. The maximum blood glucose levels (C*max*) of the CE administration group decreased by about 15% (from 229.3 ± 14.5 to 194.0 ± 7.4, *p* < 0.01) and 11% (from 225.1 ± 13.8 to 201.1 ± 7.2 hr·mg/dL, *p* < 0.05) in the sucrose and starch loading tests, respectively. Our findings suggest that citrus leaf extracts standardized to hesperidin may reduce postprandial blood glucose levels through the observed inhibitory effect against sucrase, which results in delayed carbohydrate absorption. Our findings provide a biochemical rationale for further evaluating the benefits of citrus leaves.

## 1. Introduction

According to recent statistics, the incidence of Type 2 Diabetes (T2D) is expected to increase to over 700 million people globally by 2045 [1]. T2D is characterized by increased blood glucose levels due to resistance to insulin, and these increased blood glucose levels result in metabolic abnormalities in lipid and carbohydrate metabolism [2]; it is characterized initially by resistance to insulin and later by the total or partial loss of insulin secretion, leading to chronic hyperglycemia [3,4]. T2D causes increased oxidation potential, which results in a wide variety of complications, such as retinopathy, nephropathy, and an increased risk of cardiovascular disease [5,6].

T2D is characterized by increased blood glucose levels due to the malfunction of multiple homeostatic mechanisms that manage glucose levels following the consumption of dietary carbohydrates. These carbohydrates are digested by carbohydrate-hydrolyzing enzymes, such as pancreatic α-amylase and small intestinal α-glucosidases, such as maltase, sucrase, and glucoamylase [7,8], resulting in glucose liberation. The inhibition of these enzymes has been a useful approach for managing postprandial blood glucose levels due to the delay in glucose uptake in the small intestine [3,7,9,10]. In recent years, various pharmacological approaches have been developed for the management of T2D [11], including medications that target the inhibition of carbohydrate-hydrolyzing enzymes, such as voglibose, acarbose, and miglitol [12]. Unfortunately, such medications are often accompanied by various side effects, including abdominal distension, flatulence, meteorism, liver toxicity, and possibly diarrhea [10]. For this reason, natural products that result in similar therapeutic effects have been evaluated to develop solutions without or with fewer side effects.

There is a wide variety of citrus fruits that are native to Korea, China, and Japan, and their peels have been used for many years for traditional natural medicinal purposes. Citrus fruits, such as orange, mandarin, lemon, and lime, are well known for their numerous health benefits, including radical scavenging [13] and anti-obesity effects [14]. In the Republic of Korea (South Korea), there is the mandarin (*Citrus unshiu*), which is native to Jeju Island and is the most widely consumed citrus fruit. Most mandarins are harvested when ripe; however, both the cultivation and consumption of premature fruits have increased in recent years, as unripened mandarin has been identified to contain higher amounts of organic acids, dietary fiber, flavonoids, and polyphenols [15].

Citrus fruit consumption has been well correlated with better health outcomes [16]. There are several epidemiological studies that correlate citrus fruit consumption with reductions in T2D [17], obesity [18,19], and non-fatty liver disease [20]. These health benefits have been associated with unique phenolic compounds found in citrus, such as the flavonoid compounds naringin, naringenin, hesperidin, and glycosides (hesperetin) [21]. However, little is known about the health benefits of citrus leaves for the management of T2D.

Hesperidin (HP) (Figure 1) is one of the flavonoid glycosides abundantly present in the peels of citrus fruits and contains an aglycone bonded to rutinose, a disaccharide [22]. HP exhibits a wide range of promising health benefits, including anti-diabetic activities [23], and is also known to decrease postprandial blood glucose levels by stimulating insulin sensitivity in diabetic rats [24,25]. However, the detailed mechanism of its anti-diabetic effect has not yet been thoroughly established.

Therefore, in this study, we prepared Jeju citrus leaf extracts using hot distilled water (CW) and ethyl alcohol (CE), and then we (1) identified and quantified bioactive compounds by using HPLC; (2) compared the in vitro inhibitory effects of CW, CE, and HP on α-glucosidases and α-amylase; and (3) conducted an in vivo animal study to evaluate the effects of the relevant bioactive compound (HP) and ethyl alcohol extract (CE) on postprandial hyperglycemia and compared their effects to a known nutraceutical α-glucosidase inhibitor, GO2KA1 (GO), in a sucrose- or starch-loading Sprague Dawley (SD) rat model.

## 2. Results

### 2.1. Hesperidin Content Analysis Using HPLC System

The hesperidin contents of citrus leaf extracts were analyzed using HPLC, as described in Materials and Methods (Figure 2 and Table 1). Table 1 shows the hesperidin contents of the tested water and ethyl alcohol extracts. The hesperidin content of CE was 39.17 ± 0.07 mg/g, and that of CW was 15.80 ± 0.18 mg/g-extract. In the case of citrus extracts from other parts, the highest hesperidin content was recorded in the peel (40.49 ± 0.08 mg/g), and the lowest value was recorded in the pulp (4.99 ± 0.06 mg/g).

The hesperidin content of CE was similar to that of dried citrus peel. These results indicate that underutilized citrus leaves might be a viable additional source of hesperidin. Currently, citrus leaves are considered an agricultural by-product and waste, and due to their high content of hesperidin, their utilization as a functional food ingredient can definitely be considered.

Oxidative stress is a major side effect resulting from increased glucose levels in the blood [26]. Therefore, it is essential that any dietary approaches to managing T2D include phenolic compounds, such as hesperidin, since they have a number of health benefits, such as effective antioxidant and hypoglycemic activities [27]. Citrus fruits are well-known dietary sources of phenolic compounds, especially hesperidin [28], and it has been reported that the plant variety, species, and growing/harvesting conditions play a very important role in the phenolic contents of plants [27]. Our findings suggest that the underutilized Jeju citrus leaves contain significant quantities of hesperidin, with the ethanol extract containing almost double the amount of hesperidin when compared to the water extract (Figure 2 and Table 1).

### 2.2. α-Glucosidase-Inhibitory Activity

The α-glucosidase-inhibitory activities of water (CW) and ethanol (CE) extracts of Jeju citrus leaves, hesperidin (HP), and a positive control (GO) were evaluated in a dose-dependent manner (Figure 3). The highest inhibitory effect, as determined by the IC_50_ value, was observed for HP (0.99 mg/mL), followed by CE (2.62 mg/mL) and then CW (4.99 mg/mL) (Table 2).

α-Glucosidase inhibitors reduce the enzymatic digestion of dietary carbohydrates, delaying the release of glucose and thus resulting in reduced postprandial blood glucose levels [9]. Our observations suggest that HP has a significant in vitro α-glucosidase-inhibitory effect, which probably contributes to the higher inhibitory effect of CE (when compared to CW), since CE has higher hesperidin content (Table 1 and Table 2).

### 2.3. Sucrase-Inhibitory Activity

The dose-dependent in vitro inhibitory effect of CW, CE, and HP against sucrase was also evaluated (Figure 4, Table 2). Our results suggest that HP and CE had similar inhibitory effects, as defined by their IC_50_ values (0.50 mg/mL and 0.64 mg/mL, respectively), while CW did not exert a significant inhibitory effect against sucrase (Table 2).

Rat small intestinal sucrase, maltase, and glucoamylase are key α-glucosidases that catalyze the hydrolysis of major disaccharides/oligosaccharides to glucose [29]. To understand the specificity of the observed α-glucosidase-inhibitory activity (Figure 3 and Table 2), the effect of the tested extracts and hesperidin on rat small intestinal sucrase, an enzyme responsible for sucrose hydrolysis to glucose and fructose, was examined. Our results suggest that CE has the greatest potential to reduce sucrose digestion, probably due to the higher hesperidin content (Figure 4 and Table 2). It is also interesting to note that both CE and HP resulted in inhibitory effects similar to those of the positive control GO (Table 2).

### 2.4. Maltase- and Glucoamylase-Inhibitory Activities

The dose-dependent in vitro inhibitory effects of GO, CW, CE, and HP against maltase and glucoamylase were evaluated (Figure 5 and Figure 6, Table 2). Our results suggest that HP and CE had high inhibitory activity against glucoamylase, as defined by their IC_50_ values (0.86 mg/mL and 2.03 mg/mL, respectively), while CW had low inhibitory activity (3.37 mg/mL) (Table 2).

### 2.5. α-Amylase Inhibitory Activity

α-Amylase inhibitors reduce the production of maltose from starch and eventually cause starch to behave like a dietary fiber, thus reducing the liberation of glucose and postprandial blood glucose levels [30]. The dose-dependent effect of our extracts (CW, CE, and HP) against porcine α-amylase was evaluated. Little or no inhibition was observed against porcine α-amylase (Table 2). Excessive α-amylase inhibition has been linked to side effects due to an increase in non-digested starch in the large intestine, and it has been reported that many phenolic phytochemicals are weak inhibitors of α-amylase [30].

Overall, our in vitro findings suggest that CE has greater potential to inhibit α-glucosidases and sucrase (Table 2). Inhibition of these enzymes is an important factor in controlling postprandial blood glucose levels after meals and therefore in improving postprandial hyperglycemia. To better understand the above findings, CE and HP were further evaluated using an SD rat model.

### 2.6. Blood-Glucose-Lowering Effect of Jeju Citrus Extract and Hesperidin In Vivo

To confirm the observed in vitro inhibition of sucrase activity, we evaluated the effect of two different doses of CE and HP on the glycemic response at 0, 30, 60, and 120 min after sucrose loading (2.0 g/kg body weight (b.w.)) in SD rats (Figure 7 and Figure 8).

After oral administration of CE and HP at two different doses (0.1 and 0.5 g/kg-b.w.) and sucrose (2.0 g/kg-b.w.) alone or together with one of the test compounds in SD rats, blood sugar changes were measured for 2 h (Figure 7 and Figure 8). Briefly, both treatments resulted in reduced postprandial blood glucose levels. The pharmacodynamics for the sucrose loading test can give us a better understanding of the observed results (Table 3 and Table 4). We observed that both HP and CE treatments reduced both C*max* and AUC*t* when compared to the control. It is interesting to note that the CE outcomes were not different from those of the positive control (GO) that we used (Table 3 and Table 4). Our observations confirm the observed sucrase-inhibitory effect and also confirm that hesperidin is a potential bioactive compound for the observed efficacy.

The effect of two different doses of CE and HP on the glycemic response 0, 30, 60, and 120 min after starch loading (2.0 g/kg body weight (b.w.)) in SD rats was further evaluated (Figure 9 and Figure 10). Briefly, both treatments resulted in reduced postprandial blood glucose levels. The pharmacodynamics from the starch loading test can give us a better understanding of the observed results (Table 5 and Table 6). HP treatment reduced both C*max* and AUC*t* when compared to the control (Table 5). However, CE administration did not result in a significant C*max* or AUC*t* reduction at the lowest tested dose (0.1 g/kg-b.w.), while at the highest tested dose (0.5 g/kg-b.w.), only C*max* was significantly reduced when compared to control (Table 5). This is a clear indication that CE has a very small inhibitory effect against α-amylase, as suggested by our previous in vitro observations.

Interestingly, hesperidin, which did not show α-amylase-inhibitory activity in the in vitro experiments, caused reduced blood glucose levels following starch loading in our animal model. Starch digestion is initiated by α-amylase and then finalized by the action of two α-glucosidases, namely, maltase and glucoamylase. With the above in mind, we can speculate that hesperidin might have a strong inhibitory effect against maltase or glucoamylase, and this should be evaluated in a future study.

## 3. Discussion

Jeju citrus (*Citrus unshiu*) is a popular citrus fruit native to Jeju Island in Korea. Previous reports have determined that Jeju citrus has various health benefits [4,13], which might be attributed to the significant levels of flavonoids, such as hesperidin [15]. In this study, we evaluated the hesperidin (HP) content of water and ethanol leaf extracts (CW and CE, respectively) of Jeju citrus, along with the potential blood glucose management effect, via the inhibition of carbohydrate-hydrolyzing enzymes.

We identified that indeed HP is a major flavonoid present in Jeju citrus leaf extracts, and CE had significantly higher HP content when compared to CW (39.17 mg/g and 15.80 mg/g, respectively) (Table 1). This is not a surprising finding, since hesperidin has minimal solubility in water. Previous reports have identified hesperidin in citrus leaves [31,32], but no reports on hesperidin in Jeju citrus (*Citrus unshiu*) were identified. To understand the potential hesperidin-mediated effects of CW and CE on carbohydrate-hydrolyzing enzymes, we evaluated and compared (through IC_50_ values) the inhibitory effects of both the extracts and HP against rat α-glucosidase, sucrase, and α-amylase.

All extracts demonstrated activity against rat α-glucosidase and sucrase, but the inhibitory effect against α-amylase was negligible (Table 2). CE had a superior inhibitory effect against both α-glucosidase and sucrase when compared to CW. However, for both enzymatic inhibitory activities, HP resulted in a greater inhibitory effect when compared to CE. More specifically, the IC_50_ of HP against α-glucosidase was 0.99 mg/mL, whereas the IC_50_ of CE was 2.62 mg/mL. In the case of sucrase inhibition, the IC_50_ of HP was 0.5 mg/mL, whereas the IC_50_ of CE was 0.64 mg/mL. Based on the above findings, the following conclusions can be derived: (a) ethanol extraction is the best method for developing a Jeju citrus leaf extract for potential blood glucose management via the inhibition of carbohydrate-hydrolyzing enzymes, (b) CE has the superior inhibitory effect against sucrase, (c) the observed inhibitory effect could be due to the presence of hesperidin, since HP levels in CE are much higher than those in CW, and (d) HP is an excellent inhibitor or α-glucosidase and sucrase. A recent study demonstrated that HP has a low α-glucosidase-inhibitory effect, but this study was performed using yeast-derived α-glucosidase [33]. However, there is an abundance of studies that report flavonoids, including HP, that demonstrate inhibitory effects on carbohydrate-hydrolyzing enzymes [9,17,18,21,25]. To further confirm our in vitro findings, we performed an animal study using SD rats.

In the animal study, the effect of CE and HP on the postprandial blood glucose levels of SD rats following sucrose and starch administration was evaluated. Based on the pharmacodynamic (PD) parameters, HP resulted in a significant reduction in both C*max* and AUC*t* at all tested doses (0.1 and 0.5 mg/Kg b.w.) following both sucrose and starch loading (Table 4 and Table 6). CE resulted in a significant reduction in both C*max* and AUC*t* at all tested doses (0.1 and 0.5 mg/Kg b.w.) only with sucrose loading (Table 3). In the case of the starch loading test, CE administration resulted in a significant reduction in C*max* only at the highest tested dose (0.5 mg/Kg b.w.) (Table 5). Regarding the observed results following sucrose loading, both CE and HP resulted in a significant reduction in postprandial blood glucose levels. This in vivo observation confirms our in vitro observation that suggests that both CE and HP exert very strong inhibitory effects against sucrase (Table 2). It is interesting to highlight that both CE and HP resulted in IC_50_ values very similar to that of the positive control (GO) (Table 2). Also, based on our in vitro findings, we can observe that CE has much lower inhibitory activity against rat α-glucosidase when compared to the inhibitory activity of HP (Table 2). The synthetic substrate used for this method (*p*NPG) contains an alpha 1,4-glycosidic bond, which is the one present in maltose. Once starch is digested by α-amylase, the resulting maltose is digested by small intestinal maltase to liberate two glucose molecules. Our in vivo findings suggest that the observed postprandial blood glucose level responses following starch administration correlate with our in vitro α-glucosidase inhibition observations, since HP had much higher inhibition when compared to CE (Table 2). As a result, HP resulted in significantly improved postprandial blood glucose management following starch loading at both doses, while CE administration reduced only C*max* and only at the highest tested dose (Table 5 and Table 6).

## 4. Materials and Methods

### 4.1. Materials

Jeju citrus leaf extract (*Citrus unshiu*) powders were donated by the Jeju Institute of Korean Medicine (Jejusi, Jeju, Republic of Korea). The positive control, GO2KA1^®^ (Kunpoong Bio Co. Ltd., Seoul, Republic of Korea), which is made with chitobiose, the main indicator compound, is individually recognized by the Korean Ministry of Food and Drug Safety (KFDA) as a health functional food that “helps reduce postprandial blood sugar levels” (KFDA 2018-10). Rat small intestinal acetone powders (EC 3.2.1.20), Porcine pancreatic α-amylase (EC 3.2.1.1), and soluble starch (S9765-50G) were purchased from Sigma-Aldrich Co. (St. Louis, MO, USA).

### 4.2. Sample Preparation

*C. unshiu* leaves were harvested in August 2023 in Jejusi, Jeju, Republic of Korea. The collected leaves were washed, dried using hot air at 60 °C for 24 h, and subsequently ground into a powder. The optimized conditions for the extraction of the indicator components were established by testing various conditions, including hot water, different ethanol concentrations, extraction temperatures, and extraction times, and sequential extraction, with a sample-to-solvent ratio of 1:10 (*w/v*). The pulverized samples were then extracted with 50% ethanol at a ratio of 1:10 (*w*/*v*) at 50 °C for 2 h, resulting in CE, and with a water ratio of 1:10 (*w*/*v*) at 95 °C for 1 h, resulting in CW. After extraction, the solution was concentrated at 40 °C using a vacuum rotary evaporator (BUCHI, Co., New Castle, DE, USA), followed by freeze-drying (IlshinBioBase Co., Ltd., Gyeonggi, Republic of Korea) to obtain the powder. The powder obtained was stored at −70 °C until used for experimentation.

### 4.3. Hesperidin Content Analysis Using HPLC System

A Waters 2998 HPLC system, comprising 2 pumps, a column oven, an autosampler, and a Waters 2998 photodiode array (PDA) detector, was used to identify 19 flavonoids (Waters, Milford, MA, USA). Using Empower software (version 2.0; Waters), instrument control and data acquisition were performed. Hesperidin separation was performed using an XBridge BEH C18 column (4.6 × 250 mm ID, 5 μm; Waters, Milford, MA, USA), with detection monitored at 280 nm. The mobile phase consisted of an aqueous solution of 0.1% phosphoric acid (A) and 0.1% phosphoric acid in acetonitrile (B) under the following gradient conditions: 0.0–5.0 min, 80.0% A; 5.0–40.0 min, 70.0% A; 40.0–41.0 min, 100.0% A; 41.0–44.0 min, 100.0% A; 44.0–45.0 min, 80.0% A; 45.0–50.0 min, 80.0% A. The separation was conducted at 40 °C in a column oven, with a sample injection volume of 10 μL and a flow rate of 0.8 mL/min. Stock solutions of the 19 flavonoids standard (2 mg/mL) were prepared by dissolving them in ethanol/DMSO (1/1, *v*/*v*). Standard solutions at concentrations of 25, 50, 100, and 200 μg/mL were prepared by diluting the stock solutions with ethanol/DMSO. Concentrations were verified using a standard curve, with correlation coefficients exceeding 0.999. Hesperidin in the citrus leaf extract was identified by comparing retention times and UV spectra to those of standard solutions. Hesperidin concentrations were determined by calculating the integrated peak area of each sample in relation to the corresponding standards.

### 4.4. Rat Small Intestinal α-glucosidase Inhibition Assay

To investigate the inhibitory activity of citrus leaf extracts on the absorption of glucose, rat intestinal α-glucosidase-inhibitory activity was determined using the substrate *p*-nitrophenyl-α-D-glucopyranoside (*p*NPG) according to [34] with a slight modification. Rat intestinal acetone powder (0.6 g) was dissolved in 9 mL of 0.1 M sodium phosphate buffer 0.9%, and then the suspension was sonicated 10 times for 1 min at 4 °C. After centrifugation (13,000× *g*) for 30 min at 4 °C, the recovered supernatant was used for the assay. Then, 50 μL of the sample solution and 0.1 M phosphate buffer (pH 6.9, 100 μL) containing rat intestinal α-glucosidase solution (1.0 U/mL) was incubated for 10 min at 37 °C. After the incubation, 50 μL of a *p*NPG solution (5 mM) in 0.1 M phosphate buffer (pH 6.9) was added to each well at timed intervals. The reaction mixtures were further incubated for 30 min at 37 °C. Absorbance was measured at 405 nm and compared to a control that had 50 μL of buffer solution in place of the extract using a micro-plate reader (SpectraMAx^®^ i3, Molecular devices LLC, Wals, Austria). The rat α-glucosidase-inhibitory activity is expressed as percent inhibition and was calculated as follows:$$Inhibition\left(\%\right)=\left[\frac{∆A_{control\;405}-∆A_{extract\;405}}{∆A_{control\;405}}\right]\times 100$$

### 4.5. Porcine α-Amylase Inhibition Assay

A porcine pancreatic α-amylase assay was conducted based on the method in [32] with slight modifications. First, 200 μL of the sample solution and 500 μL of 0.02 M sodium phosphate buffer (pH 6.9 with 0.006 M sodium chloride) containing an α-amylase solution (0.5 mg/mL, 15 U/mL) were incubated for 10 min at 25 °C. After incubation, 500 μL of a 1% starch solution in 0.02 M sodium phosphate buffer was added. Then, the reaction mixture was incubated for 10 min at 25 °C. The reaction was terminated with 1.0 mL of 3,5-dinitrosalicylic acid (DNS). The reaction mixture was then incubated in a boiling water bath for 5 min and then cooled to room temperature. The reaction mixture was then diluted by adding 1.0 mL of distilled water, and absorbance was measured at 540 nm with a micro-plate reader (SpectraMAx^®^ i3; Molecular Devices LLC, Wals, Austria).

### 4.6. Sucrase, Maltase, and Glucoamylase Inhibition Assay

A sucrase, maltase, and glucoamylase inhibition assay was performed by a method described in a previous study [3]. The crude enzyme solution prepared from rat intestinal acetone powder was used as the small intestinal sucrase, maltase, and glucoamylase. Rat intestinal acetone powder (600 mg) was dissolved in 9 mL of 0.1 M sodium phosphate buffer, and the suspension was sonicated 10 times for 1 min at 4 °C. After centrifugation (10,000× *g*) for 30 min at 4 °C, the recovered supernatant was used for the assay. The enzyme-inhibitory activity was measured by incubating a solution of an enzyme (100 μL), 0.1 M phosphate buffer (pH 7.0, 50 μL) containing 50 μL of 68 and 36 mg/mL sucrose and maltose, respectively, or 1% soluble starch, and a solution (50 μL) with various concentrations of the sample solution. In the reaction mixture, 200 μL of 12 N of H_2_SO_4_ was added to terminate this reaction, after which the amount of liberated glucose was measured by the glucose oxidase method. The inhibitory activity was calculated using the following formula:$$Inhibition\left(\%\right)=\left[\frac{∆A_{control\;530}-∆A_{extract\;530}}{∆A_{control\;530}}\right]\times 100$$

### 4.7. In Vivo Animal Model

Animal testing processes were approved by the Institutional Animal Care and Use Committee (IACUC) of Hannam University (Approval number: HNU2024-007). This study was reported in accordance with ARRIVE guidelines (https://arriveguidelines.org). Sprague Dawley (SD) rats (5-week-old males) were fed with a standard diet (Samyang Diet Co., Seoul, Republic of Korea) and with water ad libitum for 1 week. SD rats were used to investigate the inhibitory action of CE and HP on postprandial hyperglycemia induced by carbohydrate loading, as described in [30]. The rats were housed in a ventilated room at 25 ± 2 °C with 50 ± 7% relative humidity and under an alternating 12 h light/dark cycle. After 6 groups of 5 male SD rats (180~200 g) were fasted for 24 h, sucrose or starch at 2.0 g/kg body weight was orally administered; concurrently, a negative control group (no treatment) and a positive control group receiving a known nutraceutical α-glucosidase inhibitor, GO2KA1, were tested.

### 4.8. Blood Analysis

After administration, blood samples were taken from the tail, and blood glucose levels were measured at 0, 0.5, 1, and 2 h. Blood sugar levels were measured using the glucose oxidase method and compared with the control group. Parameters for blood sugar levels were calculated. The maximum observed peak glucose level (C*max*) and observed peak time (T*max*) were determined based on the observed data. The area under the blood glucose–time curve up to the last sampling time-point (AUC*t*) was estimated using the trapezoid rule.

### 4.9. Statistical Analysis

All data are expressed as mean ± SD. Statistical analysis was conducted using the statistical package SPSS 11 (Statistical Package for Social Science 11, SPSS Inc., Chicago, IL, USA) program, and each group was analyzed through one-way ANOVA analysis and Duncan’s test (*p* < 0.05). Additionally, statistical significance in the animal studies was determined using Student’s *t*-test (* *p* < 0.05; ** *p* < 0.01; and *** *p* < 0.001).

## 5. Conclusions

Type 2 Diabetes is a metabolic disorder characterized by elevated blood glucose levels resulting from excessive dietary carbohydrate consumption and a reduction in insulin sensitivity. It is a global health problem with a lot of microvascular complications. It is imperative to determine natural approaches to managing blood glucose levels and preventing the progression of prediabetes to Type 2 Diabetes. Our results suggest that Jeju citrus leaves are a potential source of hesperidin, which can reduce postprandial blood glucose levels. More specifically, we observed that Jeju citrus leaves extracted in ethanol contain significant quantities of hesperidin. Hesperidin is a well-studied flavonoid and has been associated with various health benefits. Our in vitro and in vivo studies suggest a biochemical rationale for further animal and clinical studies to further validate the mechanism of action and potential efficacy of the Jeju citrus ethanol leaf extract in blood glucose management. The potential mechanism for the observed blood glucose management potential involves the hesperidin-mediated inhibition of carbohydrate-hydrolyzing enzymes.

## Figures and Tables

**Figure 1 ijms-25-13721-f001:**
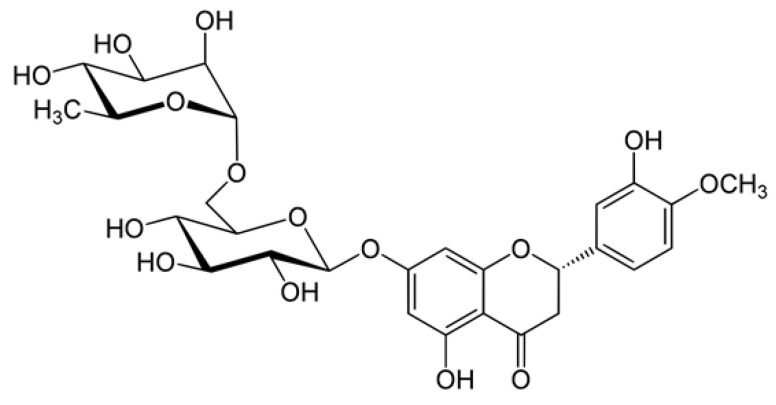
Chemical structure of hesperidin.

**Figure 2 ijms-25-13721-f002:**
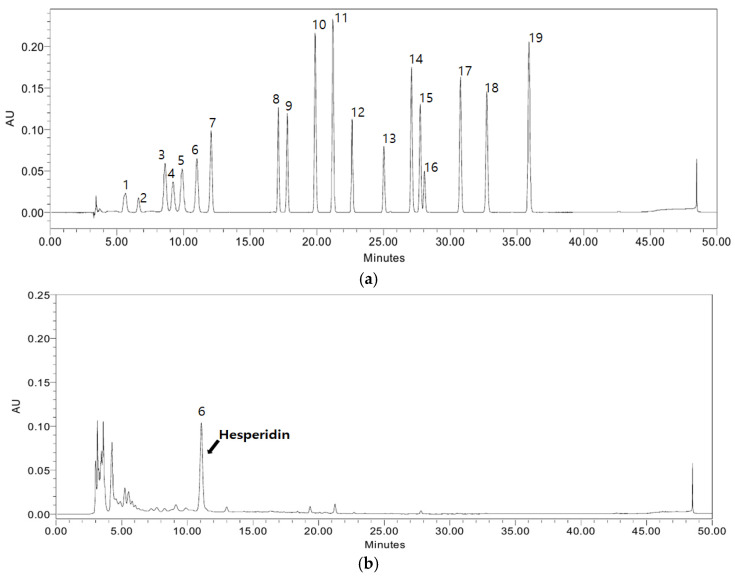
HPLC profiles of citrus leaf extracts (standard solution (**a**), hot-water extract (**b**), and ethyl alcohol extract (**c**)). 1. Rutin; 2. Neoeriocitrin; 3. Narirutin; 4. Rhoifolin; 5. Naringin; 6. Hesperidin; 7. Neohesperidin; 8. Neoponcirin; 9. Poncirin; 10. Naringenin; 11. Hesperetin; 12. Isosinensetin; 13. Sinensetin; 14. 4,5,7-Trimethoxy flavon; 15. Nobiletin; 16. 4,5,6,7-Tetramethoxy flavon; 17. Tangeretin; 18. 5-Demethyl nobiletin; and 19. Gardenin B.

**Figure 3 ijms-25-13721-f003:**
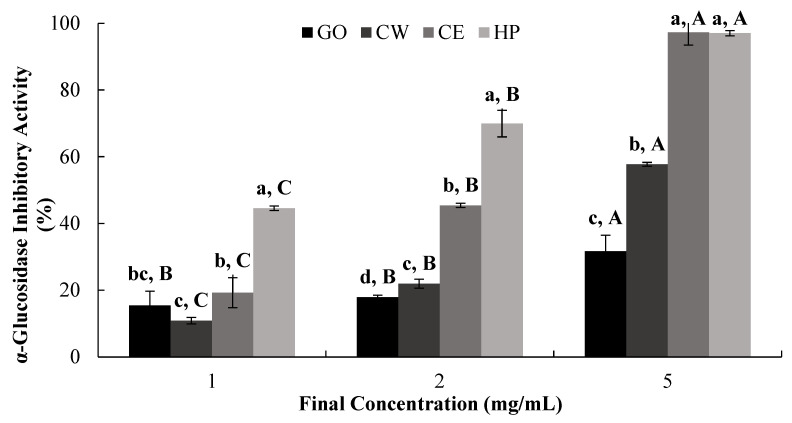
Dose-dependent changes in SD rat small intestinal α-glucosidase-inhibitory activity (% inhibition) of GO2KA1 (GO), Jeju citrus leaf hot-water extract (CW), Jeju citrus leaf ethyl alcohol extract (CE), and hesperidin (HP). Different corresponding letters indicate significant differences at *p* < 0.05 by Duncan’s test. ^a–d^ First letter indicates differences among different samples, and ^A–C^ second one indicates differences among different concentrations of same samples.

**Figure 4 ijms-25-13721-f004:**
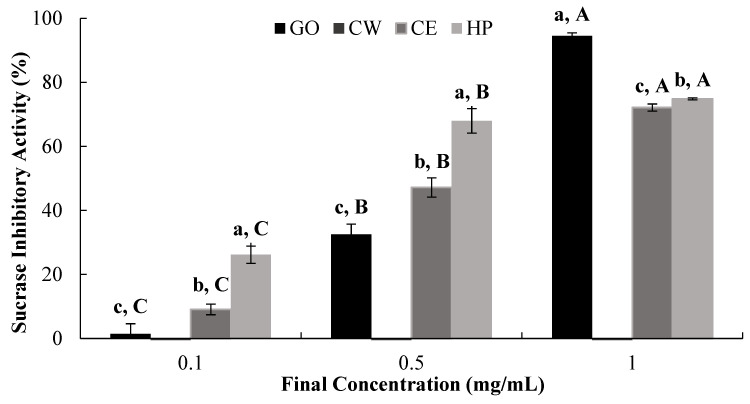
Dose-dependent changes in SD rat small intestinal sucrase-inhibitory activity (% inhibition) of GO2KA1 (GO), Jeju citrus leaf hot-water extract (CW), Jeju citrus leaf ethyl alcohol extract (CE), and hesperidin (HP). Different corresponding letters indicate significant differences at *p* < 0.05 by Duncan’s test. ^a–c^ First letter indicates differences among different samples, and ^A–C^ second one indicates differences among different concentrations of same samples.

**Figure 5 ijms-25-13721-f005:**
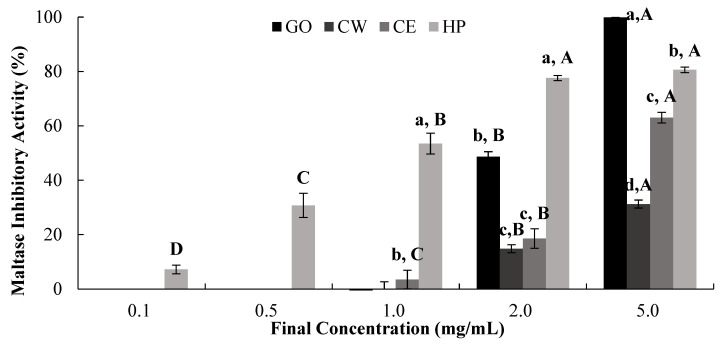
Dose-dependent changes in SD rat small intestinal maltase-inhibitory activity (% inhibition) of GO2KA1 (GO), Jeju citrus leaf hot-water extract (CW), Jeju citrus leaf ethyl alcohol extract (CE), and hesperidin (HP). Different corresponding letters indicate significant differences at *p* < 0.05 by Duncan’s test. ^a–c^ First letter indicates differences among different samples, and ^A–D^ second one indicates differences among different concentrations of same samples.

**Figure 6 ijms-25-13721-f006:**
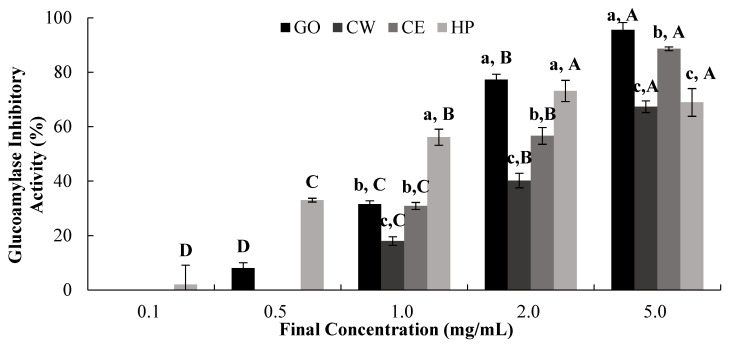
Dose-dependent changes in SD rat small intestinal glucoamylase-inhibitory activity (% inhibition) of GO2KA1 (GO), Jeju citrus leaf hot-water extract (CW), Jeju citrus leaf ethyl alcohol extract (CE), and hesperidin (HP). Different corresponding letters indicate significant differences at *p* < 0.05 by Duncan’s test. ^a–c^ First letter indicates differences among different samples, and ^A–D^ second one indicates differences among different concentrations of same samples.

**Figure 7 ijms-25-13721-f007:**
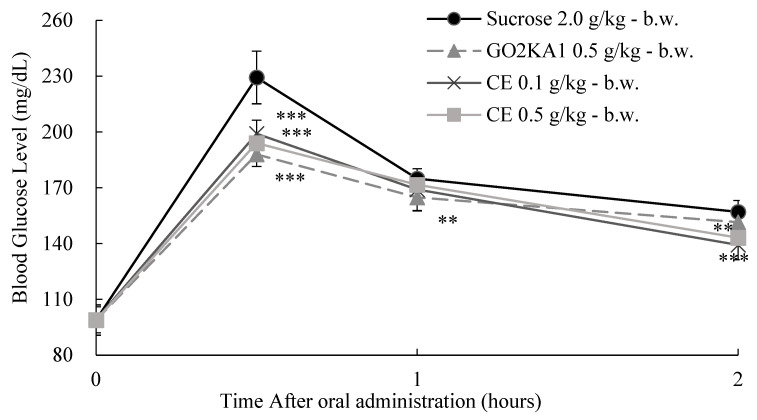
Dose-dependent anti-hyperglycemic effect of ethyl alcohol extracts of citrus leaves (CE) in sucrose loading test. After fasting for 24 h, 5-week-old male SD rats were orally administered sucrose solution (2.0 g/kg-body weight (b.w.)) with or without samples (CE 0.1 g/kg-b.w., CE 0.5 g/kg-b.w., and positive control: GO2KA1 0.5 g/kg-b.w.). Each point represents mean ± standard deviation (*n* = 10). ** *p* < 0.01 and *** *p* < 0.001 compared to different samples at the same concentration by unpaired Student’s *t*-test.

**Figure 8 ijms-25-13721-f008:**
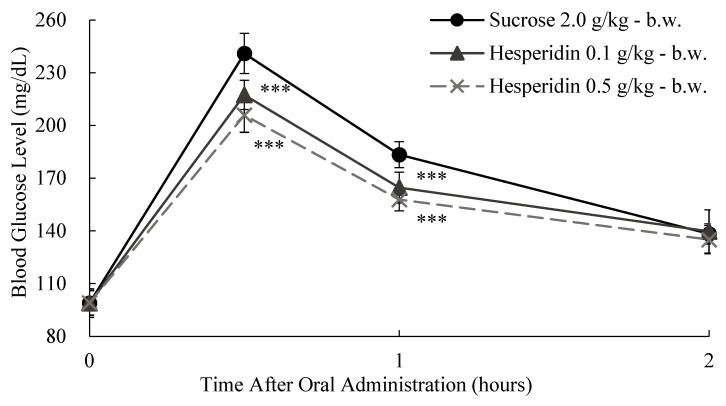
The dose-dependent anti-hyperglycemic effect of hesperidin (HP) on sucrose loading test results. After fasting for 24 h, 5-week-old male SD rats were orally administered a sucrose solution (2.0 g/kg-body weight (b.w.)) with or without the test samples (HP 0.1 g/kg-b.w. and HP 0.5 g/kg-b.w.). Each point represents mean ± standard deviation (*n* = 10). **** p* < 0.001 compared to different samples at the same concentration by unpaired Student’s *t*-test.

**Figure 9 ijms-25-13721-f009:**
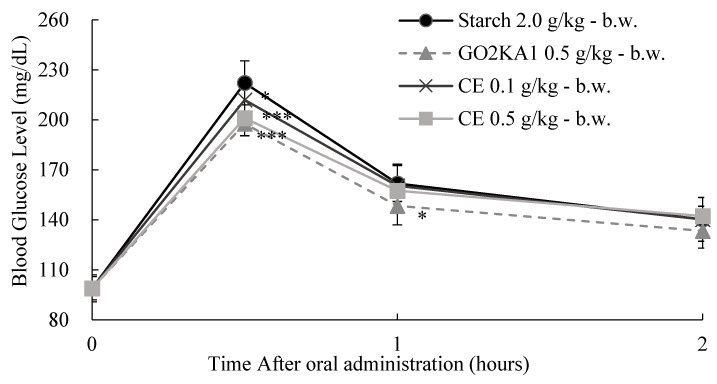
The dose-dependent anti-hyperglycemic effect of ethyl alcohol extracts of citrus leaves (CE) on starch loading test results. After fasting for 24 h, 5-week-old male SD rats were orally administered a starch solution (2.0 g/kg-body weight (b.w.)) with or without samples (CE 0.1 g/kg-b.w., CE 0.5 g/kg-b.w., and positive control: GO2KA1 0.5 g/kg-b.w.). Each point represents mean ± standard deviation (*n* = 10). ** p* < 0.05 and **** p* < 0.001 compared to different samples at the same concentration by unpaired Student’s *t*-test.

**Figure 10 ijms-25-13721-f010:**
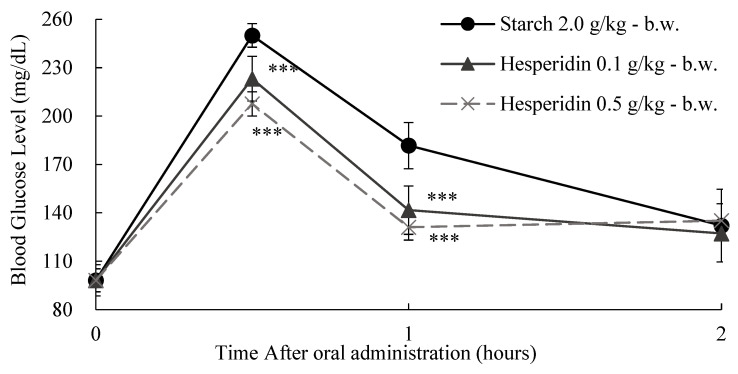
The dose-dependent anti-hyperglycemic effect of hesperidin (HP) on starch loading test results. After fasting for 24 h, 5-week-old male SD rats were orally administered a starch solution (2.0 g/kg-body weight (b.w.)) with or without samples (HP 0.1 g/kg-b.w. and HP 0.5 g/kg-b.w.). Each point represents mean ± standard deviation (*n* = 10). **** p* < 0.001 compared to different samples at the same concentration by unpaired Student’s *t*-test.

**Table 1 ijms-25-13721-t001:** Hesperidin contents (mg/g-extract) of citrus leaf hot-water (CW) and ethyl alcohol (CE) extracts.

mg/g-Extract	CW	CE
Hesperidin	15.80 ± 0.18	39.17 ± 0.07 ***

Hesperidin contents were compared between CW and CE samples by unpaired Student’s *t*-test (*** *p* < 0.001).

**Table 2 ijms-25-13721-t002:** Half-maximal inhibitory concentration (IC_50_) of citrus leaf hot-water extract and ethyl alcohol extract, hesperidin, and GO2KA1 on rat small intestinal α-glucosidase, sucrase, maltase, glucoamylase, and porcine pancreatic α-amylase activities.

			IC_50_ (mg/mL)	
	GO	CW	CE	HP
α-Glucosidase	<10.00	4.99 ± 0.07 ^a^	2.62 ± 0.06 ^b^	0.99 ± 0.14 ^c^
α-Amylase	21.04 ± 0.86	N.D.	25.63 ± 1.06 **	N.D.
Sucrase	0.58 ± 0.01 ^b^	N.D.	0.64 ± 0.02 ^a,^*	0.50 ± 0.02 ^c,^***
Maltase	2.77 ± 0.06 ^c^	7.41 ± 0.30 ^a,^**	4.12 ± 0.14 ^b,^***	1.04 ± 0.03 ^d,^***
Glucoamylase	1.41 ± 0.03 ^c^	3.37 ± 0.04 ^a,^***	2.03 ± 0.13 ^b,^*	0.86 ± 0.04 ^d,^***

The results are expressed as mean ± standard deviation. All parameters were compared between positive control and samples (GO2KA1 vs. CW, CE, and HP) by unpaired Student’s *t*-test (* *p* < 0.05, ** *p* < 0.01, and *** *p* < 0.001). ^a–d^ Different corresponding letters indicate significant differences at *p* < 0.05 by Duncan’s test.

**Table 3 ijms-25-13721-t003:** Changes in pharmacodynamic (PD) parameters of control rats and rats after administration of CE or GO2KA1 with sucrose ingestion.

Groups (g/kg-b.w.)	PD Parameters		
	C*_max_* (mg/dL)	T*_max_* (hr)	AUC*_t_* (hr·mg/dL)
Sucrose 2.0	229.3 ± 14.5 ^a^	0.5 ± 0.3	349.1 ± 10.9 ^a^
GO2KA1 0.5	188.0 ± 5.8 ^b,^**	0.5 ± 0.0	318.1 ± 8.9 ^b,^**
CE 0.1	199.1 ± 7.2 ^b,^**	0.5 ± 0.0	320.7 ± 9.7 ^b,^**
CE 0.5	194.0 ± 7.4 ^b,^**	0.5 ± 0.1	322.0 ± 11.8 ^b,^**

C*max* (maximum blood glucose levels), T*max* (time when glucose peaks), and AUC*t* (area under the curve) were observed. The results are expressed as mean ± standard deviation. All parameters were compared between control and treatment groups (GO2KA1 and CE) by unpaired Student’s *t*-test (*** p* < 0.01). ^a–b^ Different corresponding letters indicate significant differences at *p* < 0.05 by Duncan’s test.

**Table 4 ijms-25-13721-t004:** Changes in pharmacodynamic (PD) parameters of control rats and rats after administration of hesperidin (HP) with sucrose ingestion.

Groups (g/kg-b.w.)	PD Parameters		
	C*_max_* (mg/dL)	T*_max_* (hr)	AUC*_t_* (hr·mg/dL)
Sucrose 2.0	241.0 ± 11.4 ^a^	0.5 ± 0.0	351.9 ± 4.9 ^a^
HP 0.1	217.4 ± 8.0 ^b,^**	0.5 ± 0.3	326.7 ± 5.6 ^b,^***
HP 0.5	205.9 ± 9.1 ^b,^**	0.5 ± 0.3	313.6 ± 5.4 ^c,^***

C*max* (maximum blood glucose levels), T*max* (time when glucose peaks), and AUC*t* (area under the curve) were observed. The results are expressed as mean ± standard deviation. All parameters were compared between control and HP treatment groups by unpaired Student’s *t*-test (*** p* < 0.01 and **** p* < 0.001). ^a–c^ Different corresponding letters indicate significant differences at *p* < 0.05 by Duncan’s test.

**Table 5 ijms-25-13721-t005:** Changes in pharmacodynamic (PD) parameters of control rats and rats after administration of CE and GO2KA1 with starch ingestion.

Groups (g/kg-b.w.)	PD Parameters		
	C*_max_* (mg/dL)	T*_max_* (hr)	AUC*_t_* (hr·mg/dL)
Starch 2.0	225.1 ± 13.8 ^a^	0.5 ± 0.1	327.9 ± 13.9 ^a^
GO2KA1 0.5	197.7 ± 7.1 ^c,^**	0.5 ± 0.0	301.6 ± 10.4 ^b,^**
CE 0.1	212.0 ± 7.4 ^ab^	0.5 ± 0.0	321.2 ± 15.0 ^a^
CE 0.5	201.1 ± 7.2 ^bc,^*	0.5 ± 0.1	314.5 ± 8.0 ^ab^

C*max* (maximum blood glucose levels), T*max* (time when glucose peaks), and AUC*t* (area under the curve) were observed. The results are expressed as mean ± standard deviation. All parameters were compared between control and treatment groups (GO2KA1 and CE) by unpaired Student’s *t*-test (** p* < 0.05 and *** p* < 0.01). ^a–c^ Different corresponding letters indicate significant differences at *p* < 0.05 by Duncan’s test.

**Table 6 ijms-25-13721-t006:** Changes in pharmacodynamic (PD) parameters of control rats and rats after administration of hesperidin (HP) with starch ingestion.

Groups (g/kg-b.w.)	PD Parameters		
	C*_max_* (mg/dL)	T*_max_* (hr)	AUC*_t_* (hr·mg/dL)
Starch 2.0	250.0 ± 7.7 ^a^	0.5 ± 0.3	349.8 ± 18.3 ^a^
HP 0.1	223.2 ± 14.6 ^b,^*	0.5 ± 0.1	306.0 ± 17.6 ^b,^**
HP 0.5	207.6 ± 7.0 ^c,^***	0.5 ± 0.0	294.2 ± 11.1 ^b,^**

C*max* (maximum blood glucose levels), T*max* (time when glucose peaks), and AUC*t* (area under the curve) were observed. The results are expressed as mean ± standard deviation. All parameters were compared between control and treatment groups (GO2KA1 and CE) by unpaired Student’s *t*-test (** p* < 0.05, *** p* < 0.01, and **** p* < 0.001). ^a–c^ Different corresponding letters indicate significant differences at *p* < 0.05 by Duncan’s test.

## Data Availability

The datasets generated during and/or analyzed during the current study are available from the corresponding author on reasonable request.

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
