# Peer review of "Jeju Citrus (Citrus unshiu) Leaf Extract and Hesperidin Inhibit Small Intestinal α-Glucosidase Activities In Vitro and Postprandial Hyperglycemia in Animal Model"

_ijms, 2024, doi:10.3390/ijms252413721_

Round 1

Reviewer 1 Report

Comments and Suggestions for Authors

In the manuscript entitled “Jeju Citrus (Citrus unshiu) Leaf Extract and Hesperidin Inhibit 2 Small Intestinal a-Glucosidases Activities In-vitro and Post- 3 prandial Hyperglycemia in Animal Model” The authors show that citrus leaf extracts, standardized to contain hesperidin, could improve postprandial glucose peaks by inhibiting the enzyme sucrase and delaying carbohydrate absorption. These results provide a biochemical basis for the development of citrus leaf dietary supplements and for future clinical research.

The work is well written and the experiments are well thought out and well done.

Minor points

1.       The authors in the first part of the manuscript could test the inhibition of other enzymes involved in carbohydrate digestion, such as maltase and glucoamylase, to verify the hypothesis advanced on the possible role of hesperidin. Furthermore, they could perform specific in vitro tests for maltase and glucoamylase using CE and HP extracts to measure the IC50 and compare the inhibitory activity with other known inhibitors.

2.       The authors should determine whether the antioxidant activity of the extracts contributes to the reduction of hyperglycemia-induced oxidative stress. They could then measure markers of oxidative stress (e.g., malondialdehyde or reduced glutathione) in tissues of CE- and HP-treated rats during glycemic tests.

3.       The authors should also investigate whether chronic CE or HP intake affects insulin resistance and gut microbiota composition. For this purpose, they could conduct animal model studies to monitor insulin levels, HOMA-IR, and gut microbial diversity after CE or HP administration for 4-6 weeks.

4.       Finally, the authors could also better understand how hesperidin is absorbed, metabolized and distributed in the body. They could then perform in vivo pharmacokinetic studies to measure plasma levels of hesperidin after CE and HP administration.

Author Response

Reviewer 1

In the manuscript entitled “Jeju Citrus (Citrus unshiu) Leaf Extract and Hesperidin Inhibit 2 Small Intestinal a-Glucosidases Activities In-vitro and Post- 3 prandial Hyperglycemia in Animal Model” The authors show that citrus leaf extracts, standardized to contain hesperidin, could improve postprandial glucose peaks by inhibiting the enzyme sucrase and delaying carbohydrate absorption. These results provide a biochemical basis for the development of citrus leaf dietary supplements and for future clinical research.

The work is well written and the experiments are well thought out and well done.

Minor points

  1. The authors in the first part of the manuscript could test the inhibition of other enzymes involved in carbohydrate digestion, such as maltase and glucoamylase, to verify the hypothesis advanced on the possible role of hesperidin. Furthermore, they could perform specific in vitro tests for maltase and glucoamylase using CE and HP extracts to measure the IC50 and compare the inhibitory activity with other known inhibitors.

: Thank you for your important advice and comments. Following the reviewer's advice, maltase and glucoamylase inhibitory activities were measured and added (Figure 5 and 6) along with officially known alpha-glucosidase inhibitor, GO2KA1.

  1. The authors should determine whether the antioxidant activity of the extracts contributes to the reduction of hyperglycemia-induced oxidative stress. They could then measure markers of oxidative stress (e.g., malondialdehyde or reduced glutathione) in tissues of CE- and HP-treated rats during glycemic tests.

: This is correct. The antioxidant activity of phenolic phytochemicals can reduce oxidative stress caused by high blood sugar. The authors have already published numerous papers related to this topic. Therefore, in this paper, we conducted research focusing more on postprandial blood sugar suppression and submitted the results.

 The fact that it contains a large amount of hesperidin, which is well known to have antioxidant and anti-diabetic activities, means that it is already effective against diabetes and the oxidative stress caused by diabetes.

 Therefore, following the reviewer's advice, we will measure antioxidant activity and its bio-makers during future long-term animal experiments.

  1. The authors should also investigate whether chronic CE or HP intake affects insulin resistance and gut microbiota composition. For this purpose, they could conduct animal model studies to monitor insulin levels, HOMA-IR, and gut microbial diversity after CE or HP administration for 4-6 weeks.

: Thank you for your valuable advice and comments. We will make sure to reflect and verify your suggestions when conducting future research.

  1. Finally, the authors could also better understand how hesperidin is absorbed, metabolized and distributed in the body. They could then perform in vivo pharmacokinetic studies to measure plasma levels of hesperidin after CE and HP administration.

: Numerous research results have already been reported on how hesperidin is absorbed, metabolized, and distributed in the body. In this study, we investigated whether it inhibits alpha-glucosidase, which exists in the form of a membrane protein in small intestine villi before being absorbed into the body. So instead of examining the pharmacokinetics of hesperidin, we investigated the pharmacodynamics of glucose. Following your important advice, we will conduct future research on the mechanism of blood sugar control in vivo through absorption.

Thank you for your valuable advice and comment.

Reviewer 2 Report

Comments and Suggestions for Authors

Thank you very much for your interesting research. Some points must be carefully revised:

Title. “α-glucosidase” instead of “α-glucosidases”?

Abstract. Please, rephrase “…and teas are prepared with citrus peels have demonstrated health benefits”.

Abstract. (And throughout the whole manuscript). Please, avoid the use of 1st person pronouns.

Abstract. Line 23. “Inhibited” instead of “exhibited”?

Introduction. Line 37. “Recent” instead of “resent”?

Introduction.  A figure describing the structure of hesperidin might reinforce this section.

Introduction. Do you have data about hesperidin content in citrus fruits, particularly in mandarin? These data may reinforce this section.

Results. Figure 1a. Labels are unreadable.

Results. General, figures and tables: Statistical analyses? Letters or asterisks must be added to denote significant differences.

Results. Could you explain the ‘< 10.00’ result?

Discussion. Further comparisons with previously published results are missing in this section.

Materials and methods. Lines 365, 367. “w/v” instead of “g/v”?

Materials and methods. Line 370. Do you have information about the particle size of the obtained powder?

Conclusions. Specific conclusions (based on the current work results) must be included. The first part of this section is very general and cannot be considered as a conclusion per se.

Author Response

Dear Reviewer,

Thank you very much for your interesting research. Some points must be carefully revised:

Title. “α-glucosidase” instead of “α-glucosidases”?

: alpha-Glucosidases digest terminal glucose units from carbohydrates. These include, maltase, sucrase, glucoamylase and lactase. There is an alpha-glucosidase enzyme that is produced by yeast, but it his experiment we did not evaluate this specific enzyme since we used rat-intestinal powder to extract the enzyme. Therefore, we prefer to mention in the title “glucosidases”. However, if the reviewer insists, we will be happy to change to “glucosidase”.

Abstract. Please, rephrase “…and teas are prepared with citrus peels have demonstrated health benefits”.

: Yes, I agree. I have corrected it as you pointed out. (and tea made from citrus peels has demonstrated health benefits)

Abstract. (And throughout the whole manuscript). Please, avoid the use of 1st person pronouns.

: This has been addressed within the manuscript

Abstract. Line 23. “Inhibited” instead of “exhibited”?

: Yes, I agree. I have corrected it as you pointed out.

Introduction. Line 37. “Recent” instead of “resent”?

: Yes, I agree. I have corrected it as you pointed out.

Introduction.  A figure describing the structure of hesperidin might reinforce this section.

: Yes, I agree. I have added figure as you pointed out.

Introduction. Do you have data about hesperidin content in citrus fruits, particularly in mandarin? These data may reinforce this section.

: Yes, I agree. I have mentioned “hesperidin content in citrus pulp and peel” in this section as you pointed out.

Results. Figure 1a. Labels are unreadable.

: Yes, I agree. I have corrected it as you pointed out.

Results. General, figures and tables: Statistical analyses? Letters or asterisks must be added to denote significant differences.

: Yes, I agree. I have corrected it as you pointed out.

Results. Could you explain the ‘< 10.00’ result?

: This means that the IC50 value, which is the 50% inhibitory concentration, is less than 10 mg/mL.

Discussion. Further comparisons with previously published results are missing in this section.

: Based on the literature review we were not able to find any reports with Jeju Citrus (Citrus unshiu) for hesperidin content or type 2 diabetes management potential. However, we identified similar work with flavonoids and other citrus fruits, that are presented. Additions with hesperidin content in other citrus fruit leaves have been made.

Materials and methods. Lines 365, 367. “w/v” instead of “g/v”?

: Yes, I agree. I have corrected it as you pointed out.

Materials and methods. Line 370. Do you have information about the particle size of the obtained powder?

: Yes, I agree. I have corrected it as you pointed out.

Conclusions. Specific conclusions (based on the current work results) must be included. The first part of this section is very general and cannot be considered as a conclusion per se.

: Thank you for the recommendation. The conclusion has been adjusted according to the reviewer’s recommendations.

Thank you for your valuable advice and comments.

Round 2

Reviewer 2 Report

Comments and Suggestions for Authors

Thank you very much for this improved version.

However, there is still a problem with the results, since letters or asterisks were not added to Figures (3, 4, 5, 6) and Tables (1,2) to denote significant differences according to the statistical analysis.

Author Response

Dear Reviewer,

Yes, Figures (3, 4, 5, and 6) and Tables (1, 2) were revised according to reviewer's comment.

letters or asterisks were added to Figures (3, 4, 5, 6) and Tables (1,2) to denote significant differences according to the statistical analysis (Duncan's test and unpaired Student’s t-test).

Thank you for your valuable advice and comments.